# D-Dimer beyond Diagnosis of Pulmonary Embolism: Its Implication for Long-Term Prognosis in Cardio-Oncology Era

**DOI:** 10.3390/jpm13020226

**Published:** 2023-01-27

**Authors:** Masafumi Himeno, Yuji Nagatomo, Akira Miyauchi, Aimi Sakamoto, Keita Kiyose, Midori Yukino-Iwashita, Akane Kawai, Tsukasa Naganuma, Satonori Maekawara, Ayami Naito, Kazuki Kagami, Yusuke Yumita, Risako Yasuda, Takumi Toya, Yukinori Ikegami, Nobuyuki Masaki, Takeshi Adachi

**Affiliations:** Department of Cardiology, National Defense Medical College, Tokorozawa 359-8513, Japan

**Keywords:** venous thromboembolism, pulmonary embolism, malignancy, D-dimer

## Abstract

Venous thromboembolism (VTE) is a common comorbidity of cancer, often referred to as cancer-associated thrombosis (CAT). Even though its prevalence has been increasing, its clinical picture has not been thoroughly investigated. In this single-center retrospective observational study, 259 patients who were treated for pulmonary embolism (PE) between January 2015 and December 2020 were available for analysis. The patients were divided by the presence or absence of concomitant malignancy, and those with malignancy (N = 120, 46%) were further classified into active (N = 40, 15%) and inactive groups according to the treatment status of malignancy. In patients with malignancy, PE was more often diagnosed incidentally by computed tomography or D-dimer testing, and the proportion of massive PE was lower. Although D-dimer levels overall decreased after the initiation of anticoagulation therapy, concomitant malignancy was independently associated with higher D-dimer at discharge despite the lower severity of PE at onset. The patients with malignancy had a poor prognosis during post-discharge follow-up. Active malignancy was independently associated with major adverse cardiovascular events (MACE) and major bleeding. D-dimer at discharge was an independent predictor of mortality even after adjustment for malignancy. This study’s findings suggest that CAT-PE patients might have hypercoagulable states, which can potentially lead to a poorer prognosis.

## 1. Introduction

Cancer is a leading cause of death worldwide and currently, the global cancer burden is increasing and expected to dramatically rise in the future [1]. In recent years, along with advances in diagnostic modalities and the advent of new anticancer agents, the prognosis for cancer patients has been improving [2,3]. It is well known that venous thromboembolism (VTE) is a common comorbidity of cancer, often referred to as cancer-associated thrombosis (CAT) [4,5]. Thus, the prevalence of VTE as CAT has been increasing in current clinical practice [4,6]. It has also been pointed out that recurrence and bleeding events in VTE patients with malignancy are more frequent than in patients without malignancy [7]. In addition, issues related to novel anticancer agents (e.g., tyrosine kinase inhibitors), such as drug-induced thrombosis, have emerged recently [8]. In this context, there has been insufficient real-world evidence on the clinical picture of pulmonary embolism (PE) as CAT. 

D-dimer is commonly measured for the screening of VTE in clinical practice. In fact, high D-dimer levels indicate activated coagulation and fibrinolysis and are associated with the incidence of VTE in a population-based cohort study [9] and in patients with malignancies. [10,11] The prognostic impact of D-dimer has not been thoroughly investigated in patients with PE. 

Therefore, in the present study, we sought to explore the characteristics, in-hospital course including response to anticoagulation therapy, and long-term clinical outcomes of PE patients with or without malignancy in contemporary clinical practice. We also sought to explore the predictive value of D-dimer at PE onset or discharge for the long-term outcome.

## 2. Materials and Methods

### 2.1. Study Design and Subjects

The present study was conducted as a single-center retrospective observational study at the National Defense Medical College (NDMC) Hospital. We collected all cases of suspected or confirmed PE registered in the electric medical chart in NDMC Hospital between January 2015 and December 2020. After reviewing the medical records of these cases, we excluded cases with chronic thromboembolic pulmonary hypertension (CTEPH), those who did not have PE after close examination, those who had a cardiac arrest on arrival, those who had only a history of PE, and those who had been already treated at other hospitals. The cases with a confirmed diagnosis of acute PE were included in the present analysis. The cases were classified into two groups: those with malignancy and those without, based on the chart record. The patients with malignancy were further divided into two groups according to the status of malignancy as follows: the active group for patients with not-cured malignancy before or under treatment, and the inactive group for patients with malignancy already cured or in remission, which is defined as completed treatment with no signs of relapse or metastases. The status of malignancy was determined based on the chart records. The severity of PE was classified into the 3 grades based on the guidelines by [12,13,14] as follows: massive PE with hemodynamic instability, submassive PE with hemodynamically stable right heart strain findings on echocardiography and/or elevated troponin I level, and non-massive PE without them.

### 2.2. Data Collection

We collected the baseline clinical characteristics of the study subjects including demographics, medical history, comorbidities, laboratory findings at the onset of PE, and medication for the treatment of PE. The primary trigger leading to PE diagnosis was also collected from the medical chart. We evaluated the serial findings of chest computed tomography (CT) imaging as a response to the treatment for PE, and determined the thrombus regression when thrombus volume showed a reduction compared to the CT findings at PE onset, based on the radiologists’ evaluation. D-dimer data were collected at the onset of PE, and 3 and 7 days from the onset, and at discharge, and their time course changes were evaluated. Long-term follow-up data within 4 years after discharge were collected by medical chart review. The endpoints were defined as all-cause death, major adverse cardiovascular event (MACE), which consists of myocardial infarction (MI) and stroke, and major bleeding according to the definition of the International Society of Thrombosis and Haemostasis (ISTH) [15].

### 2.3. Statistical Analysis

Continuous variables are presented as mean ± standard deviation (SD) for normally distributed variables and the median (interquartile range (IQR)) for non-normally distributed data. Differences between groups were compared using the non-paired *t*-test or Mann–Whitney U rank-sum test as appropriate for unpaired data and by the chi-square test or Fisher’s exact test as appropriate for discrete variables. 

Multiple regression analysis was performed to determine independent determinants of D-dimer at discharge. The variables such as demographics, severity of PE, anticoagulation therapy at discharge, and malignancies were included as independent variables. 

Kaplan–Meier curves were drawn and the survival from clinical events such as all-cause death, recurrent PE or DVT, and major bleeding were compared between malignancy and no malignancy groups by log-rank test. These endpoints were also compared among the tertiles of D-dimer at discharge. Multivariable Cox proportional hazard model analysis was conducted to determine whether comorbid malignancy was independently associated with subsequent clinical outcomes. For all-cause death, the values were adjusted by age, sex, body mass index, serum albumin, and anticoagulation therapy at discharge. For MACE, MI + stroke, and major bleeding, the values were adjusted only by age, sex, and anticoagulation therapy at discharge due to a small number of events. Univariable Cox proportional hazard model analysis was additionally conducted for subgroups divided by malignancy status (active, inactive, and no malignancy). 

A *p* value of <0.05 was defined as statistical significance. We used JMP ver.15.2 (SAS Institute, Cary, NC, USA) for statistical analysis.

## 3. Results

### 3.1. The Patient Characteristics at Baseline

We collected all 376 cases of suspected or confirmed PE registered in the electric medical chart in NDMC Hospital between January 2015 and December 2020. After excluding 117 cases, 259 cases with a confirmed diagnosis of acute PE (age 68 (51–77) years, with male 97 (37%)) were available for analysis in the present study (Figure 1). Among the 259 PE cases, 120 patients (46%) had malignancy and the remaining 139 (54%) did not. In the malignancy group, 80 patients (31%) had an active malignancy and 40 (15%) had an inactive one (Figure 1). Regarding the breakdown of malignant tumors, gynecological and gastrointestinal tumors accounted for one-third of the total, respectively (Table 1).

The baseline characteristics of the study population according to the presence or absence of malignancy are shown in Table 2. Age was significantly higher in the malignancy group. The number of patients in the malignancy group increased in 2018–2020 compared to 2015–2017 during the study period. Regarding the predisposing factors for PE, all factors shown in Table 2 were numerically less prevalent in the malignancy group. Especially, long-term bedrest, pregnancy, and fracture were significantly less common in the malignancy group (Table 2). The other factors such as pre-existing hypertension, dyslipidemia, social history, and the percentage of patients taking antiplatelet or anticoagulant medications prior to the onset of disease did not differ significantly (Table 2).

### 3.2. The Severity of PE and the Primary Trigger Leading to PE Diagnosis

The severity of PE was compared between the malignancy group and the no-malignancy group (Figure 2A). The malignancy group showed a significantly smaller proportion of massive PE (*p* = 0.017, Figure 2A). The primary trigger leading to PE diagnosis is shown in Figure 2B. In the no-malignancy group, symptoms and signs such as hypoxia or chest pain were the primary trigger in approximately half of the patients, while in the malignancy group, laboratory findings such as D-dimer and image findings such as CT consisted of a higher percentage (73%).

### 3.3. Initial Treatment of PE and Treatment Response

Table 2 shows the medication used as an initial treatment of PE. In most cases, heparin was administered as an initial treatment both in the malignancy group and the no-malignancy group. There was no significant difference between the two groups. Figure 3A shows the proportion of patients who did not show the regression of thrombus in the CT findings during the treatment. The percentage of patients who did not show the regression of thrombus was significantly higher in the malignancy group than in the no-malignancy group (*p* = 0.028). In the active malignancy group, the percentage of patients who did not show the regression of thrombus was 12.9%, which was significantly higher than the other groups (*p* = 0.026, Figure 3A). The changes in D-dimer over time are shown in Figure 3B. D-dimer decreased after the initiation of treatment in all groups, but it was less prominent in the malignancy group, resulting in a significantly higher D-dimer value at discharge in the malignancy group (*p* < 0.001, Figure 3B). In the active malignancy group, the D-dimer value at discharge was significantly higher than in the other groups. In multiple regression analysis, concomitant malignancy or active malignancy was independently associated with a higher D-dimer value at discharge (Table 3) As an anticoagulant medication at discharge, DOAC was more commonly prescribed in the malignancy group (Table 2). The details of DOAC are shown in Appendix A Appendix A.

### 3.4. Post-Discharge Outcome

At discharge, warfarin was more commonly prescribed in the no-malignancy group, whereas DOAC was more common in the malignancy group (Table 2). During 263 (66–727) days post-discharge follow-up, 54 (21%) death, 21 (9%) MACE, 14 (6%) VTE recurrence, and 23 (10%) major bleeding occurred. Figure 4A shows the Kaplan–Meier curve for all-cause mortality within 4 years after discharge. At 4 years, all-cause mortality was as high as approximately 80% in the active malignancy group, which was extremely higher compared to the other groups (Figure 4A). Figure 4B shows the Kaplan–Meier curve for the MACE. The incidence of MACE was higher in the active group but it was not significantly different. Figure 4C shows the Kaplan–Meier curve for the composite of MI and stroke. Its incidence was significantly higher in the active group (*p* = 0.016, log-rank test), and Figure 4D shows the Kaplan–Meier curve for major bleeding events defined by ISTH criteria. [15] Its incidence was also significantly higher in the active group (*p* = 0.045, log-rank test). Figure 5 shows the Kaplan–Meier curves for all-cause death when dividing the whole study population by tertiles of D-dimer at PE onset and discharge. At PE onset, the second and third tertiles showed higher mortality compared to the first tertile (Figure 5A). At discharge, the third tertile showed the highest mortality (Figure 5B). The other endpoints such as MACE, the composite of MI and stroke, and major bleeding did not significantly differ by tertiles of D-dimer at PE onset (MACE *p* = 0.62, MI + stroke *p* = 0.71, major bleeding *p* = 0.13) or at discharge (MACE *p* = 0.40, MI + stroke *p* = 0.40, major bleeding *p* = 0.72). 

Based on these results, a multivariate Cox proportional hazard model analysis was performed for all-cause death, MACE, the composite of MI and stroke, and major bleeding events within 4 years after discharge (Table 4). Active malignancy was associated with a higher incidence of death, MACE, or the composite of MI and stroke compared to the no-malignancy group. For MACE or MI + stroke, comparison between active and inactive malignancy did not reach statistically significant differences. The incidence of major bleeding was significantly higher in active malignancy compared to inactive or no malignancy. 

Of note, the D-dimer value at discharge was a significant predictor for all-cause death, whereas D-dimer at PE onset did not remain significant after adjustment for covariates (Table 4). When the population was divided by malignancy status, D-dimer at discharge was associated with long-term mortality in active and inactive malignancy groups, but not in the no-malignancy group. There was a significant interaction between D-dimer at discharge and malignancy status (Table 5, *p* for interaction = 0.002).

Figure 6 shows the Kaplan–Meier curve for all-cause death in patients with malignancy according to the years of PE diagnosis. Compared to the patients registered between 2015 and 2016, the survival rate was more favorable for patients who were registered in the more recent period (Figure 6).

## 4. Discussion

In the present study, we reported the following main findings. First, the predisposing factors for PE were overall less prevalent in the malignancy group. This may suggest that patients with malignancies are more likely to develop VTE even in the absence of other thrombogenic factors. Second, unexpectedly, the severity of PE was milder in the malignancy group. Incidental PE detected by laboratory or image findings was more prevalent as the primary trigger leading to PE diagnosis in the malignancy group. From these findings, we speculated that PE might be detected in patients with malignancy in an earlier stage than those without malignancy, and that is why the disease was less severe at diagnosis. Third, notably, the active malignancy group showed a higher percentage of patients who did not show the regression of thrombus and a less favorable reduction in D-dimer from PE onset toward discharge, suggesting a prothrombotic phenotype in patients with malignancy. Fourth, not only all-cause death, but also the composite of MI and stroke and major bleeding was more common in the active malignancy group. Lastly, D-dimer at discharge was an independent predictor of long-term all-cause death even after adjusting for malignancy, whereas D-dimer at PE onset did remain a significant predictor after adjustment for covariates. Especially, D-dimer at discharge was associated with mortality in active and inactive malignancy groups. From these findings, we concluded that PE patients concomitant with active malignancy showed features related to hypercoagulable states, and might be more prone to both thrombotic and hemorrhagic events, and of note, they showed extremely poor prognosis after PE diagnosis. D-dimer at discharge was an informative marker for long-term mortality, especially in patients with malignancy.

### 4.1. CAT-PE in Contemporary Clinical Practice

The population in the present study has some distinct features compared to previous studies. The percentage of concomitant malignancy in VTE largely differs in the reported studies. While the increasing prevalence of VTE as CAT has been recognized [4,6], the present study reported a relatively high prevalence (46%) of concomitant malignancy compared to previous studies, including large registries (2–28%) [16,17,18,19,20,21,22,23].

In the present study, PE was less severe in patients with malignancy, which was explained by a higher percentage of incidental PE detected by D-dimer or CT scan. From these findings, PE might be detected in patients with malignancy in an earlier stage than those without malignancy, and that is why the disease was less severe at diagnosis. This may be because the malignancy group underwent regular checkups of D-dimer and CT scans during follow-up, to check the progression and/or recurrence of malignancy and the comorbidities. Thus, patients with malignancy are more likely to have a chance to detect incidental PE.

In the present study, DOAC was more commonly used compared to the recent large registries [18,19,21]. Moreover, the use of DOAC was more frequent in patients with malignancy, which reconciles with the observations from the RE-COVERY DVT/PE, a large multicenter international observational study [24]. This may be partly because the use of DOAC showed superiority to warfarin and non-inferiority to low molecular weight heparin (LMWH) in terms of subsequent clinical outcomes [25,26,27]. In addition, DOAC may be preferred since in real-world anticoagulation therapy, it is sometimes delivered by oncologists who are not used to handling warfarin. 

Approximately a decade ago, a previous nationwide survey reported that the long-term survival of VTE patients without cancer did not significantly differ compared to the general population [28] More recently, among VTE patients, those with malignancy showed poor prognosis compared to patients with unprovoked and provoked VTE in the Framingham Heart Study [29] Consistent with this finding, not surprisingly, the prognosis of patients with malignancy or active malignancy was considerably worse than the no-malignancy group in the present study. On the other hand, a number of studies have shown that CAT is a frequently seen and serious complication in patients with malignancy. Comorbid VTE was associated with higher mortality among cancer patients [30] These findings suggest thrombosis is a marker of advanced disease for patients with malignancy.

The relatively low recurrence of VTE in the present study was noted compared to the previous studies. [16,31] Although its reason is unknown, it might be due to the extremely high mortality in the malignancy group. Moreover, this might be explained by the superiority of DOACs in CAT in terms of their efficacy in VTE treatment and prevention. Its property of immediate effect or easy handling (e.g., no need for blood monitoring or dose adjustment) might be advantageous. 

Malignancy is associated not only with VTE but ATE [32], which was first shown by Trousseau [33]. In the present study, the active malignancy was an independent determinant of MACE, MI/stroke, and major bleeding events even after adjusting for anticoagulant at discharge. These findings are in line with these previous studies [5,7,19]; although, VTE recurrence did not significantly differ in the present study. From these findings, we need to pay attention to the risk of ATE and bleeding, not only VTE recurrence, in patients with CAT-PE. 

### 4.2. Time Course Changes and Prognostic Significance of D-Dimer

D-dimer, a degradation product of cross-linked fibrin, is elevated in VTE and correlated with PE severity [34]. More than a decade ago, D-dimer at PE onset was reported to be associated with acute-phase (15–30 days) [35,36], mid-term (3 months) [36,37], and long-term mortality [38]. The combination of D-dimer with the other biomarkers has also been reported to predict VTE recurrence or mortality [23,39,40]. However, according to a recent report, the predictive value of D-dimer at onset for long-term mortality might be questionable [34].

D-dimer is also elevated in various conditions such as inflammation, surgery, and cancer [10,11,38]. Although cancer was associated with higher D-dimer levels in a report more than a decade ago [38], in the present study, using contemporary real-world data, D-dimer at PE onset was not higher in the malignancy group, possibly due to the lower severity of PE. Nevertheless, the residual elevation of D-dimer at discharge in the malignancy group was noted. These findings suggest that patients with malignancy or active malignancy have hypercoagulable states and are more resistant to anticoagulation therapy. To our knowledge, this is the first study that has compared the time course changes of D-dimer between patients with malignancy and those without.

Furthermore, in the present study, D-dimer at discharge discriminated long-term mortality even after adjustment for covariates, but D-dimer at PE onset did not. Residual D-dimer elevation even under anticoagulation might more precisely reflect an intrinsic hypercoagulable state than that at PE onset, which is possibly derived from the comorbid diseases including malignancies. Accumulating preclinical data suggest that the activation of coagulation promotes tumor growth and angiogenesis, further supporting the association of the clinical hypercoagulable state and adverse cancer prognosis [41,42]. An elevated D-dimer level has been shown to predict mortality in malignancy patients without VTE. [30] Thus, the D-dimer level at discharge, rather than that at PE onset, can be employed as an informative prognostic marker in patients with PE, especially in the presence of malignancy. So far, the predictive value of D-dimer during anticoagulant therapy or after its cessation has been explored mostly for VTE recurrence [43,44]. Although one study showed the association of elevated D-dimer at discharge with VTE recurrence [16], the prognostic values of D-dimer at discharge for long-term mortality in patients with PE have been rarely reported. 

### 4.3. Limitations

The present study has several limitations. This study is a single-center retrospective study that enrolled a relatively small number of patients. In addition, in current clinical practice, DOACs are becoming the mainstream anticoagulation therapy, but warfarin use was relatively common during this study period. LMWH has been shown to be superior to warfarin in safety and efficacy for the treatment of CAT and is recommended by the latest guidelines [45,46]. However, since LMWH is not approved for the treatment of PE in Japan [12], it was not used in this study population. 

## 5. Conclusions

The findings of the present study showed CAT-PE patients had a poor prognosis, and D-dimer at discharge can be a prognostic marker for PE patients irrespective of the presence of malignancy. This study’s findings suggest CAT-PE patients might have hypercoagulable states, which can potentially lead to a poorer prognosis. Our data provide useful information on the management of PE patients, especially for those with malignancy. Along with advances in cancer treatment, it is highly likely that their prognosis will further improve and the management of CAT will become increasingly important in the future. 

## Figures and Tables

**Figure 1 jpm-13-00226-f001:**
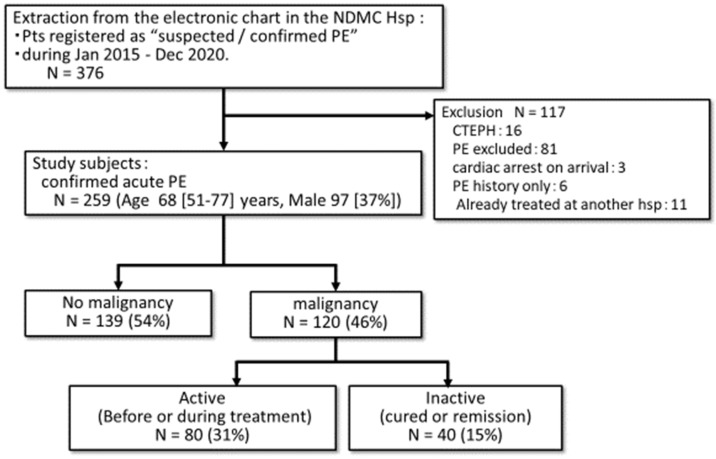
Study flowchart. The electrical chart record was reviewed and the patients who were registered as suspected or confirmed PE were extracted. Among them, those who had confirmed acute PE were available for analysis in the present study. The study subjects were divided into those who had malignancy and those who did not. PE, pulmonary embolism; CTEPH, chronic thromboembolic pulmonary hypertension; NDMC, National Defense Medical College; hsp, hospital.

**Figure 2 jpm-13-00226-f002:**
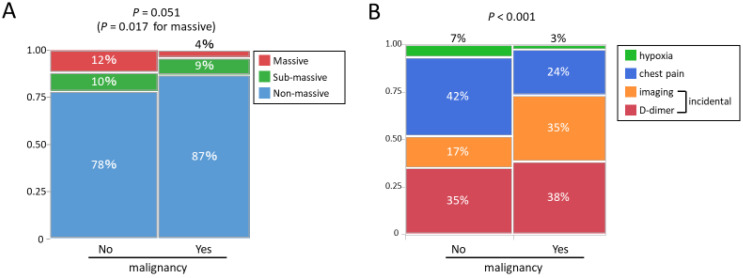
(**A**) The severity of PE and (**B**) the primary trigger leading to the diagnosis of PE: (**A**) The malignancy group showed a significantly smaller proportion of massive PE. (**B**) In the no-malignancy group, symptoms and signs such as hypoxia or chest pain were the primary trigger in approximately half of the patients, while in the malignancy group, laboratory findings such as D-dimer and image findings such as CT consisted of a higher percentage. PE, pulmonary embolism; CT, computed tomography.

**Figure 3 jpm-13-00226-f003:**
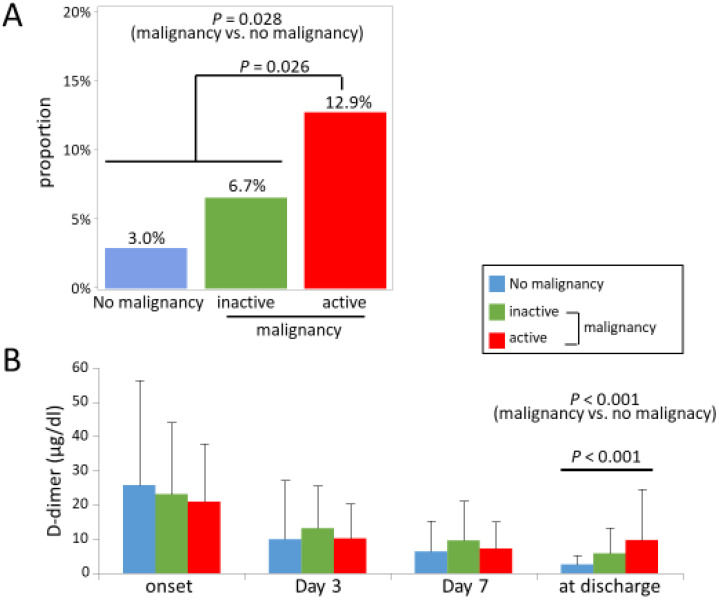
(**A**) The percentage of patients who did not show the regression of thrombus on CT and (**B**) the time course changes of D-dimer during the treatment. (**A**) The percentage of patients who did not show the regression of thrombus was significantly higher in the malignancy group than in the no-malignancy group. CT, computed tomography.

**Figure 4 jpm-13-00226-f004:**
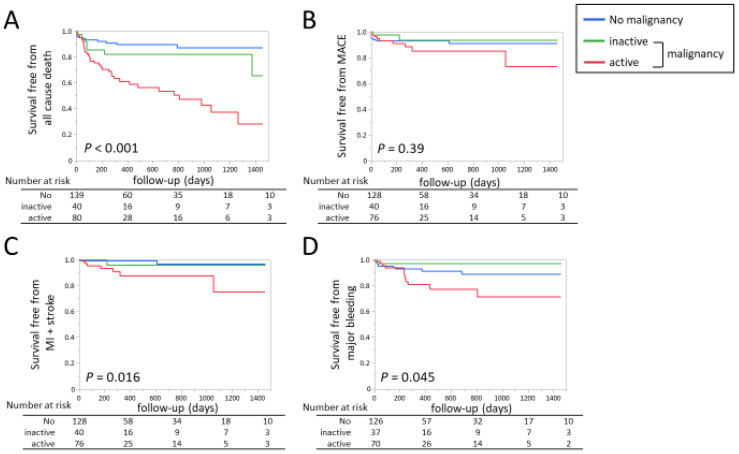
Kaplan–Meier curves for (**A**) all-cause death; (**B**) MACE defined as the composite of cardiovascular death, MI, and stroke; (**C**) the composite of MI and stroke; and (**D**) major bleeding in patients with malignancy and those without. MACE, major adverse cardiovascular events; MI, myocardial infarction.

**Figure 5 jpm-13-00226-f005:**
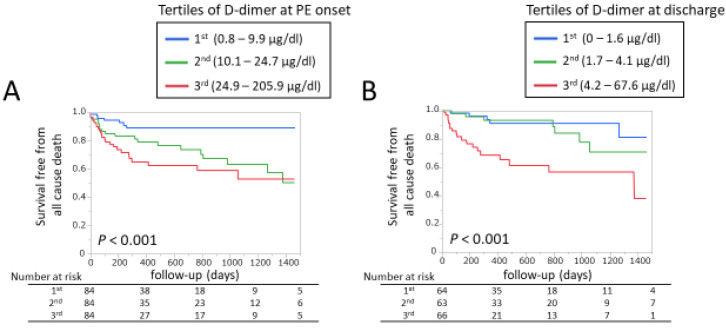
Kaplan–Meier curves for all-cause death in the whole study population divided by (**A**) tertiles of D-dimer at PE onset and (**B**) at discharge. PE, pulmonary embolism.

**Figure 6 jpm-13-00226-f006:**
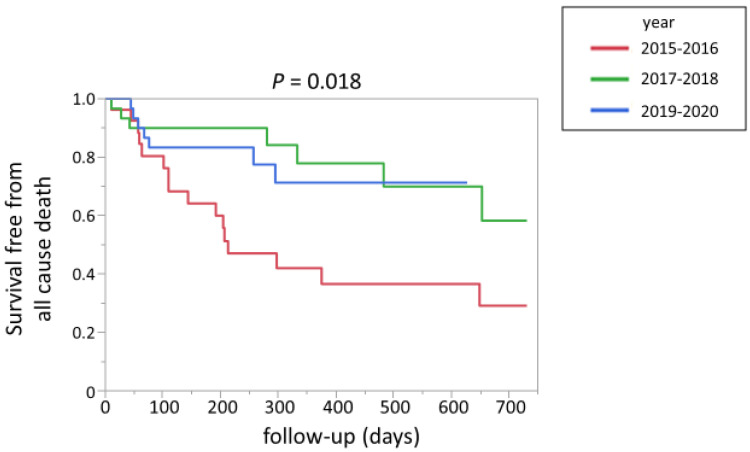
Kaplan–Meier curve for all-cause death in patients with malignancy according to the years of PE diagnosis. PE, pulmonary embolism.

**Table 1 jpm-13-00226-t001:** The details of malignancies.

		Patients with Malignancies (N = 120)
Primary Site of Malignancy		
	ovary	24 (20%)
	colon	21 (17%)
	stomach	12 (10%)
	uterus	10 (8.3%)
	lung	9 (7.5%)
	lymphocyte (lymphoma)	7 (5.8%)
	mammary gland	6 (5.0%)
	pancreas	5 (4.2%)
	bladder	4 (3.3%)
	prostate	3 (2.5%)
	liver	3 (2.5%)
	unknown	3 (2.5%)
	kidney	2 (1.6%)
	skin	2 (1.6%)
	duodenum	2 (1.6%)
	adipose tissue	1 (0.8%)
	anus	1 (0.8%)
	bone marrow (myeloma)	1 (0.8%)
	brain	1 (0.8%)
	larynx	1 (0.8%)
	pharynx	1 (0.8%)
	retroperitoneum	1 (0.8%)
**Chemotherapy**		43 (36%)
**Radiation**		3 (2.5%)

**Table 2 jpm-13-00226-t002:** The baseline characteristics of the study population.

	Malignancy		
	No (N = 139)	Yes (N = 120)	*p*
age (years)	62 (42–77)	71 (61–78)	0.001
sex (male, %)	45 (32%)	52 (43%)	0.069
BMI	22.7 (21.0–25.6)	22.8 (20.0–25.2)	0.36
BSA	1.59 (1.47–1.76)	1.58 (1.44–1.71)	0.36
Triggers of PE			
bed rest	34 (24%)	8 (7%)	<0.001
post-operation	25 (18%)	18 (15%)	0.52
pregnancy	17 (12%)	0 (0%)	<0.001
infection	12 (9%)	8 (7%)	0.55
collagen disease	8 (6%)	2 (2%)	0.076
bone fracture	8 (6%)	0 (0%)	0.008
congenital disease	2 (1.4%)	0 (0%)	0.50
drug-induced	9 (6%)	4 (3%)	0.27
Comorbidities/Medical history			
hypertension	51 (37%)	42 (35%)	0.78
dyslipidemia	24 (17%)	15 (13%)	0.28
diabetes mellitus	19 (14%)	15 (13%)	0.78
hemodialysis	1 (1%)	0 (0%)	1.0
stroke	11 (8%)	8 (7%)	0.70
venous thromboembolism	11 (8%)	10 (8%)	0.90
arterial thromboembolism	3 (2%)	2 (2%)	1.0
Social history			
tobacco (never/ex/current)	97/25/17 (70%/18%/12%)	65/33/22 (54%/28%/18%)	0.035
alcohol	49 (35%)	42 (35%)	0.97
Medication			
antiplatelet	8 (6%)	6 (5%)	0.78
anticoagulant	12 (9%)	7 (6%)	0.37
Initial treatment			0.59
heparin	111 (83%)	96 (80%)	
DOAC	20 (15%)	21 (18%)	
others	3 (2%)	3 (3%)	
Treatment at discharge			<0.001
warfarin	39 (30%)	5 (5%)	
DOAC	79 (61%)	91 (82%)	
none	12 (9%)	15 (13%)	
In-hospital death	9 (6%)	8 (7%)	0.95

BMI, body mass index; BSA, body surface area; PE, pulmonary embolism; DOAC direct oral anti-coagulant.

**Table 3 jpm-13-00226-t003:** Multiple regression analysis for D-dimer level at discharge.

	Model 1				Model 2			
	β	SEM	*t*-Value	*p* Value	β	SEM	*t*-Value	*p* Value
age	−0.033	0.051	−0.39	0.70	−0.006	0.050	−0.07	0.95
sex (female)	−0.076	0.80	−0.97	0.33	−0.060	0.79	−0.78	0.44
BMI	−0.019	0.18	−0.24	0.81	−0.018	0.18	−0.24	0.81
PE severity								
massive/non-massive	−0.090	2.27	−0.74	0.46	−0.093	2.25	−0.77	0.44
sub-massive/non-massive	0.067	1.92	0.56	0.57	0.076	1.90	0.65	0.52
Treatment at discharge								
DOACs/None	−0.053	1.21	−0.71	0.48	−0.022	1.20	−0.30	0.76
DOACr/None	−0.12	1.43	−1.43	0.16	−0.14	1.42	−1.72	0.09
Warfarin/None	0.005	1.76	0.06	0.95	−0.010	1.71	−0.12	0.91
Malignancy status								
malignancy (yes)	0.32	0.79	3.95	<0.001	-	-	-	-
active malignancy (yes)	-	-	-	-	0.35	0.79	4.51	<0.001

SEM, standard error of the mean; BMI, body mass index; PE, pulmonary embolism; DOAC, direct oral anticoagulant.

**Table 4 jpm-13-00226-t004:** Multivariable Cox proportional hazard model analysis for clinical endpoints.

			All Cause Death									
			Model 1				Model 2					
			HR	95% CI	*p*		HR		95% CI		*p*	
D-dimer at PE onset			1.01	0.996–1.03	0.14							
D-dimer at discharge							1.06		1.03–1.09		<0.001	
Anticoagulation therapy at discharge												
DOACs/None			0.31	0.11–0.88	0.028		0.11		0.027–0.46		0.003	
DOACr/None			0.19	0.063–0.55	0.002		0.081		0.017–0.39		0.002	
Warfarin/None			0.61	0.15–2.47	0.49		0.16		0.027–0.99		0.049	
Malignancy status												
(active/no)			23.8	6.79–83.6	<0.001		9.32		2.50–34.7		<0.001	
(active/inactive)			5.30	1.92–14.6	0.001		4.17		1.16–15.0		0.028	
	**MACE**				**MI + Stroke**				**Major Bleeding**			
	**HR**	**95% CI**		** *p* **	**HR**	**95% CI**		** *p* **	**HR**	**95% CI**		** *p* **
Anticoagulation therapy at discharge												
DOACs/None	0.15	0.030–0.72		0.018	0.48	0.051–4.54		0.52	0.44	0.12–1.66		0.23
DOACr/None	0.025	0.003–0.19		<0.001	0.086	0.006–1.18		0.066	0.18	0.040–0.82		0.027
Warfarin/None	0.074	0.006–0.89		0.041	0.27	0.015–5.10		0.38	1.02	0.25–4.21		0.98
Malignancy status												
(active/no)	17.0	3.09–93.7		0.001	9.52	1.72–52.6		0.010	3.39	1.34–8.60		0.010
(active/inactive)	8.05	0.95–68.1		0.056	4.89	0.56–42.8		0.15	8.15	1.04–63.8		0.046

MACE was defined as the composite of cardiovascular death, nonfatal MI, and stroke. For all-cause death, the values were further adjusted by age, sex, body mass index, and serum albumin. For MACE, MI + stroke, and major bleeding, the values were further adjusted by age and sex. PE, pulmonary embolism; MACE, major adverse cardiac events; MI, myocardial infarction; M, male; F, female; DOAC, direct oral anticoagulant. DOACs, standard dose of DOAC; DOACr, reduced dose of DOAC.

**Table 5 jpm-13-00226-t005:** Univariable Cox proportional hazard model analysis for all-cause death in the malignancy subgroups.

	Malignancy Status								
	No			Inactive			Active		
	HR	95% CI	*p*	HR	95% CI	*p*	HR	95% CI	*p*
D-dimer at discharge	0.63	0.21–1.19	0.20	1.67	1.18–3.74	<0.001	1.05	1.03–1.08	<0.001

*p* for interaction (malignancy groups D-dimer) = 0.002.

## Data Availability

This manuscript reports the results of clinical research. The data will not be shared.

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
