# Peer review of "D-Dimer beyond Diagnosis of Pulmonary Embolism: Its Implication for Long-Term Prognosis in Cardio-Oncology Era"

_jpm, 2023, doi:10.3390/jpm13020226_

Round 1
Reviewer 1 Report
I would like to thank for the opportunity to review an interesting article by Himeno and co-authors "D-dimer beyond Diagnosis of Pulmonary Embolism: Its Implication for Pathophysiology and Long-term Prognosis in Cardio-Oncology Era". In this manuscript, the authors presented a well-designed clinical study that evaluated the clinical and prognostic value of D-dimer in patients with Pulmonary Embolism depending on the presence or absence of oncological pathology. As a result, the authors obtained a certain set of new scientific facts that are important both for research in this area and for practical medicine.
However, when reviewing the manuscript, I had questions and comments that I would like to receive answers from the authors.
1. In my opinion, the article lacks a pathophysiological aspect. In any case, based on the determination of only one biomarker in dynamics, it is difficult to make assumptions about pathophysiological mechanisms. Therefore, I would remove the references to pathophysiology in the title of the article and change one of the subsections (4.2. Pathophysiological insights from D-dimer ...) in the "Discussion" section.
2. In general, subsection 4.2. badly written. At the beginning, it repeats (lines 268-270) the information from the Introduction section (lines 42-44). The results of the study (lines 272-288) that were already presented earlier in the Results section are then re-presented. And very briefly, previous studies using D-dimer as a prognostic factor are presented. I think this section of the discussion should be revised with the addition of consideration of relevant publications (for example, references 1-3)
3. The article contains many references to publications older than 10 years (13 out of 30). It is probably the right of the authors to use such references to substantiate the study and discuss its results. But the title of the manuscript contains the words "CardioOncology Era". Apparently, this era has come not so long ago, since the authors decided to emphasize this. Therefore, literary references must, apparently, also belong to this era, or the authors must indicate how this era ("CardioOncology") differs from earlier studies conducted in another, previous era.
4. The authors state that "This is the first study which explored the prognostic values of D-dimer at discharge for the risk stratification in patients with PE" (lines 295-297). It is difficult to agree with this, since, for example, in the article by Wang et al (3), the prognostic role of D-dimer in hospital discharge was studied.
References:
1. Liu X, Zheng L, Han J, Song L, Geng H, Liu Y. Joint analysis of D-dimer, N-terminal pro b-type natriuretic peptide, and cardiac troponin I on predicting acute pulmonary embolism relapse and mortality. Sci Rep. 2021 Jul 21;11(1):14909. doi: 10.1038/s41598-021-94346-7.
2. Maestre A, Trujillo-Santos J, Visoná A, Lobo JL, Grau E, Malý R, Duce R, Monreal M; RIETE Investigators. D-dimer levels and 90-day outcome in patients with acute pulmonary embolism with or without cancer. Thromb Res. 2014 Mar;133(3):384-9. doi: 10.1016/j.thromres.2013.12.044.
3. Wang Y, Liu ZH, Zhang HL, Luo Q, Zhao ZH, Zhao Q. Predictive value of D-dimer test for recurrent venous thromboembolism at hospital discharge in patients with acute pulmonary embolism. J Thromb Thrombolysis. 2011 Nov;32(4):410-6. doi: 10.1007/s11239-011-0625-2.
Reviewer 2 Report
Thanks to the authors for this interesting article on PE in cancer patients however I do not agree with the authors in their conclusion that d-dimer is a infomative prognostic marker in patients with PE
Methods - there are sections in the methods that should be included in the results section, including figure 1. The authors define PE severity but do not define what is inactive cancer in remission.
results - page 5 line 135-137 would be more appropriate in the conclusion rather than the results. In page 6 the authors comment of regression of thrombus but I cannot find anything in methods about how this was assessed.
I am unclear what patients are included in figure 5. Is it all patients then divided by tertiles of Ddimer or just the malignancy patients?
It would be interesting to have more information on the anticoagulants use and the outcome of patients depending on which anticoagulant was used.
Conclusion - the authors comment that raised d-dminer on discharge was associated with worse outcome but this is likely because d-dimer is raised in patients with cancer as cancer is a pro-thrombotic state. Consequently, I do not think d-dimer is useful in prognosis in patients with cancer
Round 2
Reviewer 1 Report
I am grateful to the authors for the work done to correct the manuscript and answers to my questions, I am quite satisfied. I have no other questions or comments.